# Immune Checkpoint Inhibitors in Mismatch Repair Proficient/Microsatellite Stable Metastatic Colorectal Cancer Patients: Insights from the AtezoTRIBE and MAYA Trials

**DOI:** 10.3390/cancers14010052

**Published:** 2021-12-23

**Authors:** Marco Maria Germani, Roberto Moretto

**Affiliations:** 1Unit of Medical Oncology 2, Azienda Ospedaliero-Universitaria Pisana, Via Roma, 67, 56126 Pisa, Italy; m.germani@studenti.unipi.it; 2Department of Translational Research and New Technologies in Medicine and Surgery, University of Pisa, 56126 Pisa, Italy

**Keywords:** metastatic colorectal cancer, proficient mismatch repair system/microsatellite stability, immune checkpoint inhibitors, FOLFOXIRI/bevacizumab, temozolomide

## Abstract

**Simple Summary:**

Immune-checkpoint inhibitors (ICI) show modest activity and efficacy in microsatellite stable (MSS) metastatic colorectal cancer (mCRC) patients harbouring a proficient mismatch repair system (pMMR). Recently, two phase 2 trials -AtezoTRIBE and MAYA- have challenged this dogma through the administration of an intense first-line chemotherapy backbone consisting of FOLFOXIRI/bevacizumab in patients unselected for their microsatellite status, and immune priming with temozolomide in chemorefractory pMMR/MSS patients with silencing of O6-methylguanine-DNA methyltransferase (*MGMT*), respectively, reporting promising results. We here present the founding biological rationale of these two studies and their main findings. At the same time, we stress their strengths and drawbacks and open questions still to be address in future clinical investigations.

**Abstract:**

In metastatic colorectal cancer (mCRC), remarkable advances have been achieved with immune checkpoint inhibitors (ICIs) targeting PD-1/PD-L1 and CTLA-4, only in a small subset of tumours (4–5%), harbouring a deficient mismatch repair system (dMMR)/microsatellite instability–high (MSI-H) or mutations in the catalytic subunit of polymerase epsilon (*POLE*). Within this framework, several combination strategies have been investigated to sensitize proficient mismatch repair (pMMR)/microsatellite stable (MSS) mCRC to ICIs, with disappointing results so far. However, at the last ESMO meeting, two phase II trials AtezoTRIBE and MAYA provided promising results in this field. In the comparative AtezoTRIBE trial, the addition of atezolizumab to FOLFOXIRI (5-fluoruracil, oxaliplatin and irinotecan) and bevacizumab led to a significant advantage in terms of progression free survival (PFS) in a population of untreated mCRC patients, not selected according to MMR/MSI status. In the single-arm MAYA trial, immune priming with temozolomide in pMMR/MSS chemo-resistant mCRC patients with silencing of O6-methylguanine-DNA methyltransferase (*MGMT*) allowed reporting signals of sensitivity to the subsequent therapy with nivolumab and a low dose of ipilimumab in some patients. Here, we discuss the rationale, results, criticisms and research perspectives opened by these two studies.

## 1. Introduction

Immune checkpoint inhibitors (ICIs) targeting the programmed cell death 1 (PD1) and its ligand 1 (PD-L1) and the cytotoxic T-lymphocyte antigen-4 (CTLA-4) have revolutionized the treatment and prognosis of several solid and haematological malignancies, including lung, kidney, and urothelial cancers and melanoma and lymphoma [1,2]. However, ICIs remain largely ineffective in metastatic colorectal cancer (mCRC), where the benefit of immunotherapy is currently limited to a small subset of patients (4–5%) harbouring a deficient mismatch repair system (dMMR)/microsatellite instability–high (MSI-H) [3,4]. Several prospective phase II trials reported outstanding results for dMMR/MSI-H chemo-resistant mCRC patients treated with different anti-PD1 agents (pembrolizumab, nivolumab and dostarlimab) or with the combination of nivolumab and the anti-CTLA-4 ipilimumab [5,6,7,8].

Recently, the prospective phase III Keynote-177 trial randomizing 307 dMMR/MSI-H mCRC patients to pembrolizumab or standard-of-care chemotherapy as first-line treatment, reported longer progression-free survival (PFS) (median 16.5 versus 8.2 months; HR: 0.59, 95% CI: 0.45–0.79; *p* = 0.0002) and a trend towards better overall survival (OS) (median not reached versus 36.7 months; HR: 0.74, 95% CI: 0.53–1.03; *p* = 0.0359)—the co-primary endpoints of the study—for patients treated with immunotherapy. Notably, after a median follow up of 44.5 months, 44% of patients in the pembrolizumab arm had not progressed yet. Additionally, the other secondary endpoints, including overall response rate (ORR), second PFS, safety profile and quality of life (QoL), were in favour of the anti-PD1 agent [9,10,11]. Therefore, pembrolizumab has become the new standard of care and obtained approval from FDA and EMA in the first-line treatment of dMMR/MSI-H mCRC [12,13].

Meanwhile, the prospective single-arm phase II study CheckMate-142 showed remarkable and durable responses in 45 dMMR/MSI-H chemo-naïve mCRC patients receiving nivolumab plus low-dose ipilimumab, with 74% of patients not progressed after a median follow-up of 24.2 months and with an acceptable safety profile [8].

Results from the phase III CheckMate-8HW trial—comparing nivolumab plus low-dose ipilimumab versus nivolumab alone versus standard-of-care—and the phase III COMMIT study—comparing the anti-PD-L1 atezolizumab alone or in combination with FOLFOX (5-fluorouracil and oxaliplatin) and bevacizumab as upfront treatment of dMMR/MSI-H mCRC patients—are expected [14,15].

Recently, a prospective phase II trial reported high activity of nivolumab in six chemo-refractory mCRC patients harbouring mutations of the gene encoding for the catalytic subunit of polymerase epsilon (*POLE*) [16]. Considering the low percentage of *POLE*-mutated mCRC patients (<1%), these results may be enough for the approval of nivolumab in this small subgroup of patients.

The sensitivity to ICIs detected in dMMR/MSI-H and *POLE* mutated tumours relies on a common hypermutated phenotype that causes the consequent development of neoantigens boosting an immune-inflamed (or “hot”) microenvironment, where Cytotoxic T-lymphocytes (CTLs) activity against tumour cells is restrained by immune checkpoint, like PD1/PD-L1 and CTLA-4/B7 axes, and may be released by ICIs [17,18]. On the other hand, the majority of mCRC patients, characterized by a proficient mismatch repair system/microsatellite stable (pMMR/MSS) and *POLE* integrity (95%), has an immune-desert or immune-excluded (or “cold”) microenvironment, finally resulting in a blunted immune activation of tumour microenvironment that causes the futility of ICIs in these patients [19]. To this regard, in chemorefractory pMMR/MSS mCRC patients, Le et al. and Chen et al. showed a lack of efficacy of the anti-PD1 pembrolizumab and a modest clinical benefit of the anti-PD-L1 durvalumab plus the anti-CTLA-4 tremelimumab, reserved only to patients with a tumour mutational burden (TMB) more than 28 mut/Mb on circulating tumour DNA, respectively [4,20]. Drawing from these considerations, a growing amount of research has recently explored different combination strategies in which ICIs have been incorporated with chemotherapy, radiotherapy and biologic agents with the purpose of reshaping the microenvironment of pMMR/MSS tumours towards an immune-inflamed/hot phenotype, that may lead to ICIs sensitivity. Nevertheless, the results for these promising approaches have been largely disappointing so far [19].

At the last ESMO meeting, two phase II studies named AtezoTRIBE and MAYA, assessing combinations of ICIs with chemotherapy, have rekindled hope for the use of immunotherapy in pMMR/MSS mCRC patients, representing a major breakthrough and a promising ground for future investigations in this setting [21,22].

## 2. AtezoTRIBE Trial

AtezoTRIBE (Table 1) is a phase II multicentre, open-label, comparative trial where 218 unresectable and chemo-naïve mCRC patients, irrespective of microsatellite status, were randomized in a 1:2 ratio to receive FOLFOXIRI (5-fluorouracil, oxaliplatin and irinotecan)/bevacizumab (standard arm) or the same treatment with the addition of atezolizumab (experimental arm). Both treatments were administered up to 8 cycles, followed by 5-fluorouracil plus bevacizumab, with or without atezolizumab, according to randomized arm until progression disease, unacceptable adverse events or consent withdrawal. The primary endpoint was PFS with a 1-sided alpha error of 0.10 and a power of 85%.

The rationale of this study moves from the immunogenic cell death induced by chemotherapy and the subsequent release of neoantigens. Next, they are recognized by dendritic cells (DCs) able to present them to CTLs and, therefore, to activate immune response against cancer cells [19]. The inhibition of VEGF/VEGFR pathway by bevacizumab induces vasculature normalization that allows an increasing in CTL tumour infiltration. In addition, anti-angiogenic agents stimulate the maturation of DCs and reduce the expansion of T-reg lymphocytes and myeloid-derived suppressor cells, thus contributing to immune effector cells activation [19]. Overall, these mechanisms could modify the immune microenvironment towards an immunogenic status and thus cause the efficacy of ICIs. Indeed, a preliminary phase Ib study assessing the combination of FOLFOX/bevacizumab and the anti-PD-L1 atezolizumab showed an acceptable safety profile and an interesting activity, as well as some signals of immune activation, including the increase in activated infiltrating CTLs [23]. However, two randomized phase II trials assessing the addition of atezolizumab to fluoropyrimidine and bevacizumab compared fluoropyrimidine and bevacizumab in advanced lines, and, in the maintenance setting, reported disappointing results [24,25]. Despite these discouraging data, the AtezoTRIBE study explored whether sensitivity to ICIs could be achieved boosting the cytotoxic effect through intensification of the chemo-backbone up to the triplet FOLFOXIRI, and by using this regimen in previously untreated patients. This approach is supported by the well-established higher activity and efficacy of FOLFOXIRI/bevacizumab, with respect to the doublets FOLFOX or FOLFIRI/bevacizumab, corroborated by a recent meta-analysis, including five phase II and III randomized trials [26,27,28].

Overall, no unexpected adverse events (AEs) from the addition of atezolizumab to FOLFOXIRI and bevacizumab were reported in both the safety run-in group including the first six patients randomized in the experimental arm and in the whole safety population. Indeed, no increase in chemo- and bevacizumab-related AEs were observed and only a higher incidence of expected immune-related AEs was reported in the atezolizumab arm. After a median follow-up of 20.6 months, the primary endpoint was met with a median PFS of 13.1 and 11.5 months in the experimental and control arms, respectively (HR: 0.69, 80% CI: 0.56–0.85; *p* = 0.012). No differences were shown in terms of the ORR and R0 resection rates. OS data were not mature yet. The PFS benefit from the addition of atezolizumab was retained in all subgroups. As expected, the small subgroup of dMMR/MSI-H patients (*N* = 13) showed a remarkable advantage from immunotherapy (median PFS not reached versus 6.6 months; HR: 0.11, 80% CI: 0.04–0.35; *p* = 0.002), with only 2 events of disease progression reported, without any case of primary resistance detected in the experimental group, and with a positive interaction test of 0.010. In the pMMR/MSS cohort, a mitigated but still significant advantage was retained in the experimental arm (median PFS: 12.9 versus 11.5 months; HR: 0.78, 80% CI: 0.62–0.97; *p* = 0.071).

In summary, the AtezoTRIBE trial is the first study reporting a significant clinical benefit from the addition of ICI to chemotherapy and bevacizumab in mCRC patients. This strategy seems of particular interest in dMMR/MSI-H patients where 15–30% showed primary resistance to immunotherapy with anti-PD1 in combination with anti-CTLA-4 or alone, with respect to the 0% reported in the experimental arm of the AtezoTRIBE study. To this regard, results of the COMMIT study are expected to confirm this finding.

In the pMMR/MSS group, an advantage of only 1.5 months was shown in median PFS and, albeit a plateau of the survival curve was observed after 20 months, a longer follow-up and OS data are awaited to clearly delineate the magnitude of clinical benefit of atezolizumab addition to FOLFOXIRI and bevacizumab. Moreover, considering the possibility of pseudo-progression in patients treated with immunotherapy and the absence of placebo in the experimental arm, investigators may have been induced to continue treatment in case of progression only in patients treated with the experimental arm; therefore, the benefit of the addition of atezolizumab to FOLFOXIRI/bevacizumab may be lower than reported [29].

The PFS advantage achieved in the experimental arm in absence of a higher ORR may be explained by the biological mechanisms of action of ICIs. In fact, atezolizumab does not cooperate with chemotherapy in tumour response, but delays progression in patients achieving tumour shrinkage with subsequent release of neoantigens and immune activation of tumour microenvironment that allows ICIs efficacy.

The limited advantage reported in pMMR/MSS population strengthens the importance of the planned translational analyses to identify potential predictive biomarkers of response to immunotherapy, including the assessment of immunoscore, tumour-infiltrating lymphocytes (TILs), TMB, homologous recombination status and others.

Waiting for data from the prospective single-arm phase II study NIVACOR, investigating the combination of FOLFOXIRI, bevacizumab and nivolumab in *RAS* or *BRAF* mutated pMMR/MSS patients in the first-line setting of mCRC, AtezoTRIBE results would need a confirmation in a phase III study in untreated pMMR/MSS mCRC patients to validate the therapeutic potential of the chemotherapy, antiangiogenic and anti-PD1/PD-L1 combination [30]. The identification of a subpopulation of special interest, based on an immunological biomarker, would be highly welcome.

## 3. MAYA Trial

MAYA (Table 1) is a prospective single-arm phase II study that enrolled chemo-resistant mCRC patients with centrally confirmed MSS status, MGMT silencing assessed by promoter methylation of *O6-methylguanine-DNA methyl-transferase* (*MGMT*) gene and complete immunohistochemistry loss of MGMT protein. Participants are to receive two cycles of temozolomide (TMZ) (phase I), followed by the addition of nivolumab and low-dose of ipilimumab (phase II), only in cases of disease control achieved during phase I. The primary endpoint was the 8-month PFS rate in patients entered in phase II of the study.

The design of the MAYA trial is based on an intriguing biological rationale. Briefly, *MGMT* is a gene implicated in the repair of DNA alterations caused by alkylating agents, including TMZ. When *MGMT* is inactivated by hypermethylation of its promoter, sensitivity to TMZ is enhanced [31,32,33,34]. Moreover, retrospective data showed that sensitivity to TMZ was mainly restricted to pMMR/MSS tumours with complete MGMT protein loss detected with immunohistochemistry as showed in studies assessing the efficacy of TMZ alone or in combination with other chemotherapeutic agents, such as capecitabine and irinotecan [34,35,36]. After initial disease response, secondary resistance to TMZ can be due to a *MGMT* re-expression or selection of MGMT expressing sub-clones, or to an hypermutated status derived by acquired mutations in MMR genes that can sensitize mCRC to ICIs [37].

In the MAYA trial, 204 out of 703 (29%) screened patients were deemed molecularly eligible. Overall, 142 out of 703 (19%) were enrolled in the phase I and only 33, corresponding to <5% of the initially 703 screened patients, proceeded to phase II.

No new safety signals emerged from the two phases of the study in terms of G3/4 TMZ- and ICI-related AEs. With 12 out of 33 patients (36%) progression-free after 8 months, the MAYA trial met its primary endpoint. In addition, promising results were observed in terms of ORR (42%) and median OS (18.5 months).

These results should be interpreted with caution due to the lack of a control arm assessing the efficacy of TMZ monotherapy, which prevented the investigators to discern the effect of the addition of immunotherapy, with respect to TMZ alone. Indeed, considering that not all patients initially sensitive to TMZ develop a hypermutated phenotype, only a subgroup of patients may benefit from the addition of immunotherapy. However, future analyses differentiating the ORR reported during the phase I of the trial (with TMZ) from the ORR reported in the phase II (with TMZ plus ipilimumab plus nivolumab) along with ongoing translational analyses, could mitigate this limitation. Moreover, another ongoing single-arm phase II study named ARETHUSA—conducted in the same setting of MAYA trial and assessing the effect of pembrolizumab administered in patients that develop a TMB exceeding 20 mutations/Megabase after disease control and subsequent progression to TMZ—may clarify the benefit of the TMZ and ICIs combinations [38].

In summary, the results of MAYA trial put an emphasis on the potentials of pre-clinical research to translate our advances in cancer cell biology into a clinical benefit for heavily pre-treated patients. However, a larger randomized study for confirming these results might be difficult to realize due to the demanding molecular selection required.

## 4. Conclusions

The promising data from the AtezoTRIBE and MAYA trials break off a long period of stagnation and disappointing results in the landscape of immunotherapy in pMMR/MSS mCRC, though a more mature follow-up and the extensive publishing of overall results are awaited to solidify these findings. The intensification of chemotherapy in the first-line setting, and the TMZ administration in *MGMT-*silenced chemo-refractory patients, seems able to sensitize immune-desert/cold mCRCs to immunotherapy, possibly rewiring an inflamed/hot tumour microenvironment—finally unleashed against the tumour by ICIs. However, larger confirmatory and translational studies are needed to identify those patients who achieved more benefit from these strategies.

## Figures and Tables

**Table 1 cancers-14-00052-t001:** Main characteristic of AtezoTRIBE and MAYA trials.

Trial Characteristics	AtezoTRIBE	MAYA
Study design	Randomized Phase II comparative trial	Single arm Phase II trial
Clinical selection	Chemo-naïve unresectable mCRC ECOG-PS = 0/1 aged < 70 years or 70–75 years with ECOG-PS = 0	Pre-treated patients progressed to fluoropyrimidine, oxaliplatin and irinotecan (+anti-EGFR in *RAS/BRAF* wild-type tumours)
Molecular selection	pMMR/MSS and dMMR/MSI-H patients	pMMR/MSS patients with *MGMT* promoter methylation by pyrosequencing and MGMT complete loss by immunohistochemistry
Intervention	Eight cycles of induction with FOLFOXIRI + bevacizumab ± atezolizumab followed by 5-FU + bevacizumab ± atezolizumab until progression disease, unacceptable adverse events or consent withdrawal	Two cycles of temozolomide (phase I) followed by the addition of nivolumab + low-dose ipilimumab (phase II) in case of disease control during phase I until progression disease, unacceptable adverse events or consent withdrawal
Endpoint	PFS	8-month PFS rate
Planned statistical significance	1-sided alpha error: 0.10 beta error: 0.15	Fleming single-stage design: p0 (null hypothesis) = 5% p1 (alternative hypothesis) = 20% At least 4 patients out of 27 patients PFS at 8 months for rejecting null hypothesis
Patient enrolled	218	142 (phase I), 33 (phase II)
Median PFS	ITT population: experimental arm: 13.1 months control arm: 11.5 months (HR: 0.69, 80%CI: 0.56–0.85; *p* = 0.012) dMMR/MSI-H population: experimental arm: not reached control arm: 6.6 months (HR: 0.11, 80%CI: 0.04–0.35; *p* = 0.002) pMMR/MSS population: experimental arm: 12.9 months control arm: 11.5 (HR: 0.78, 80%CI: 0.62–0.97; *p* = 0.071)	7.1 months (8 months PFS rate: 12/33)
Median OS	Not available	18.5 months
ORR	experimental arm: 59% control arm: 64% (*p* = 0.175)	42%

Legend: ITT—intention to treat; mCRC—metastatic colorectal cancer; ORR—overall response rate; OS—overall survival; PFS—progression-free survival; pMMR/MSS—proficient mismatch repair/microsatellite stable; dMMR/MSI-H—deficient mismatch repair/microsatellite high.

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
