# Peer review of "Immune Checkpoint Inhibitors in Mismatch Repair Proficient/Microsatellite Stable Metastatic Colorectal Cancer Patients: Insights from the AtezoTRIBE and MAYA Trials"

_cancers, 2021, doi:10.3390/cancers14010052_

Round 1
Reviewer 1 Report
The AtezoTribe study initiates an immune response by inducing cell death to release neoantigens that dendritic cells can pick up and present to CTLS. VEGF/VEGFR inhibitor normalizes vasculature and enhances CTL tumor penetration using the chemical cocktail FOLFOXIRI and bevacizumab. The experimental arm of the study was given the above cocktail in combination with atezolizumab, an ICI inhibitor of PD1. The treatment overall enhanced patient survival from 11.5 to 13.5 months. The authors call into question the study’s conclusions and whether the introduction of ICI enhances treatment. Specifically, they believe that a pseudo-progression resulted in the deviation of the experimental group to more conventional treatments, which can explain the differences between PFS and expected ORR. This is further supported by the lack of enhanced treatment in the pMMR/MSS patients.
The MAYA trial was conducted in chemoresistant mCRC patients with silencing of MGMT. Patients were treated with a combination of the PD1 and CTLA-4 Inhibitor with the chemotherapeutic temozolomide. The study focused on the enhanced sensitivity to temozolomide to MGMT silencing. In the study, 12 out of 33 patients were progression-free after eight months. The study showed an ORR of 42% and OS of 18.5 months. The authors critic the study and its lack of a control temozolomide group.
The manuscript does an excellent job of summarizing the current studies that are investigating the combination of conventional therapeutics and ICI for the treatment of mCRC. Furthermore, the authors bring up important critiques of the two studies. I especially agree with the significance of the AtezoTribe study which had only a modest increase in patient survival and a lack of a control group of the MAYA trial. In both cases, the positive results could be inflated or misrepresented. Furthermore, the AtezoTribe study was heavily dependent on the cytotoxicity of the chemotherapeutic cocktail and increasing the dose in the experimental group.
One major critic of the paper is that the studies are still ongoing or are currently being replicated which means that the overall result could change following publication. Furthermore, I feel the review could benefit from a brief review of previous trials with just the single treatments to give a better understanding of the clinical significance of the combination therapy. With this being said I feel that the comments on the study are valid and should be seriously considered by those conducted subsequent trials. Furthermore, the trials being presented represent an exciting new avenue for therapeutic intervention and possibility of a cooperative approach for dealing with difficult to treat tumors.
Author Response
Thank you for your suggestions: we provided a brief report including the already known activity and efficacy of FOLFOXIRI+bevacizumab, TMZ monotherapy and ICI in pMMR mCRC. We also stressed that extensive data publishing are still awaited to strengthen these findings.
Reviewer 2 Report
Nice article to stress the new found potential of ICI in colorectal cancer.
Author Response
Thank you for your comment
Reviewer 3 Report
This is a timely report in an area of interest in colorectal cancer treatment. Only minor comments:
- title should be more specific. 'new perspectives' is a bit misleading and made me expect new ideas would be proposed in the commentary. This was primarily a review of two clinical trials. Suggested title could be: 'Immune checkpoint inhibitors in mismatch repair proficient / microsatellite stable metastatic colorectal cancer patients: insights from the AtezoTRIBE and MAYA Trials
- line 79 - 'sensibility' should read 'sensitivity'
- AtezoTRIBE trial - should better explain exactly what atezolizumab is
- in general English is very good and manuscript is well-written. Some sentences are far too long though - eg. line 100
Author Response
Thank you for your comments, we modified the manuscript accordingly.